# Spatial Distribution Characteristics and Influencing Factors of Traditional Villages in Guangxi Zhuang Autonomous Region

**Meiyan Li, Wen Ouyang * and Dayu Zhang**

School of Architecture and Urban Planning, Beijing University of Civil Engineering and Architecture, Beijing 100044, China
* Correspondence: ouyangwen@bucea.edu.cn; Tel.: +86-010-139110062852

**Abstract:** This study comprises 669 national and local traditional villages in the Guangxi Zhuang Autonomous Region. Using the ArcGIS software platform, the nearest neighbor index, coefficient of variation, and kernel density tools are used to describe the distribution density characteristics of traditional villages; the imbalance index method and the Gini coefficient are used to describe the equilibrium index of the distribution of traditional villages in municipalities and geographical subdivisions. This study demonstrates that Guangxi's traditional villages are spatially distributed with "one main and two vices". Traditional villages are unevenly concentrated in Guilin City and the northern parts of Hezhou and Liuzhou. They are geographically concentrated in the Yuecheng Ling mountain range and Guibei's surrounding flat areas, Guizhong's Shengtang Mountain range, and the Guidong's alluvial river plains. Traditional villages are more prevalent in mountainous areas, and their construction and development take the water resources of rivers and flood protection into account. The research results of this paper have an important guiding significance for considering the internal rules of the spatial distribution of traditional villages in Guangxi, so as to provide some data support for the protection planning of traditional villages in Guangxi.

**Keywords:** Guangxi; traditional villages; spatial distribution; ArcGIS; kernel density; imbalance index

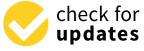



## 1. Introduction

Traditional villages are important carriers of Chinese farming culture and cultural heritage resources, with significantly high historical, cultural, scientific, artistic, economic, and social values [1] which are irreproducible. Certain issues have arisen in recent years and, due to a lack of attention to traditional villages and inappropriate protection and utilization, including the destruction of traditional texture, the destruction of traditional architectural styles, the disappearance of traditional regional cultural characteristics, and other problems, more and more ancient villages are facing the dilemma of disappearing and being destroyed. Data from a group at the China Ancient Villages Conference in 2017 showed that, in the past 15 years, the number of traditional villages in China has decreased by nearly 920,000, and that the number is decreasing at a rate of 1.6 per day. Since 2012, 5 batches of traditional Chinese villages have been selected, including 6819 villages and towns. Concurrently, several provinces, cities, and districts have accelerated the assessment and protection of traditional villages at different levels. Scientific research on the protection, revitalization, and utilization of traditional villages has continued to intensify in order to serve the science of traditional village protection by national and local governments. This mainly involves interdisciplinary fields, such as architecture, sociology, history, planning, and psychology, which have become critical in supporting the field of traditional village protection. Domestic scholars' research on traditional villages majorly focuses on traditional villages' connotations and culture [1,2], their spatial form [3,4], their spatial structure [5–7], their public space [8], and their tourism development [9,10]. The research on the spatial distribution of traditional villages and its influencing factors using geography combined

with big data analysis has become a new hot spot with the gradual popularization of the application of big data and information technology in the field of urban and rural planning and architecture [11–14]. Furthermore, most current related studies are based on core density, geographic agglomeration, and other perspectives in order to cut into the multi-perspective analysis.

Relevant policies and the literature research highlight some pressing issues in current research on traditional villages. The selected five batches of traditional national villages serve as the research object in the majority of studies. Few studies focus on the listed traditional villages at provincial, municipal, and district levels, which are also vital in terms of their historical and cultural values, as carriers of locally inherited traditional cultural characteristics and can provide the advantage of concentrated contiguous traditional villages. In order to provide scientific reference for the protection and living utilization of traditional villages in the Guangxi Zhuang Autonomous Region, this study conducts a general study of national and autonomous traditional villages in Guangxi, analyzes the spatial distribution characteristics of traditional villages based on the ArcGIS software platform and comprehensive application of various spatial analysis methods, analyzes the influence of topography and geomorphology, and rivers and water systems, and also considers other factors on the spatial distribution of traditional villages.

## 2. Data Sources and Research Methods Numerical Methodology

### 2.1. Data Sources

This study's data objects primarily include map vectors of the entire Guangxi area, information about traditional villages, and point coordinates of traditional villages in Guangxi, as well as a topographic elevation map and water system map of the entire Guangxi area.

The map vector is obtained from the Baidu map sector. The point coordinates of traditional villages are crawled by the data of the Baidu map's open platform, and the coordinates of these villages are positioned with the vector map of Guangxi through ArcGIS. The spatial distribution map of traditional villages in Guangxi is shown in Figure 1. Moreover, the information about traditional villages is more comprehensive. As of June 2019, a total of 280 traditional villages in Guangxi are included in the list of 5 batches of traditional Chinese villages announced by the Ministries of Housing and Construction, Culture and Tourism, and Finance. Furthermore, 655 traditional villages are included in the list of 3 batches of district (provincial) level traditional villages announced by the Guangxi Zhuang Autonomous Region, excluding the information of duplicate villages at the national and regional levels, to obtain a total of 669 traditional villages at the national and provincial levels in Guangxi.

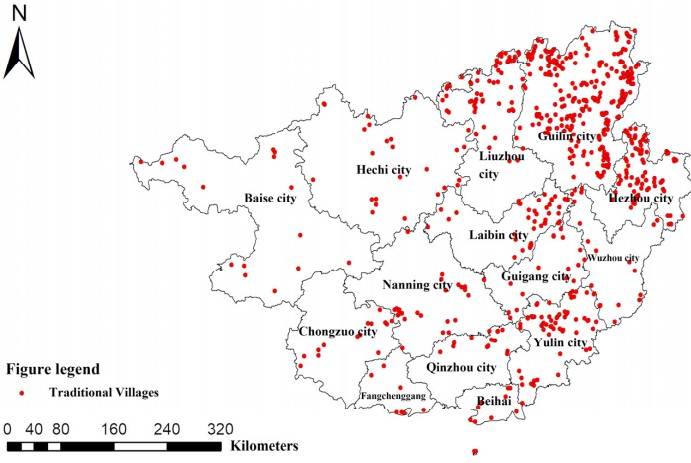

**Figure 1.** Spatial distribution of traditional villages in Guangxi.

*2.2. Methods of Research*

This study uses 669 traditional villages in the Guangxi Zhuang Autonomous Region as its research object. It abstracts their locations into point-like elements and employs the nearest neighbor index and coefficient of variation analysis to investigate the different types of spatial distribution of traditional villages in the Guangxi Province. The distribution characteristics of 669 traditional villages in the spatial scope of the whole province can then be revealed. The imbalance index was used to characterize the balanced spatial distribution of traditional villages and was analyzed in the scope of each city in Guangxi. Moreover, kernel density analysis was used to characterize the density characteristics of the spatial distribution. The ArcGIS software was used to characterize the distribution characteristics of traditional villages in Guangxi's five major geographical sub-regions, as well as the spatial distribution characteristics of traditional villages in Guangxi. The characteristics of the spatial distribution of traditional villages in each city of Guangxi were analyzed; the spatial distribution density characteristics were studied using the kernel density analysis method; the Gini coefficient was used to characterize the distribution characteristics of traditional villages in Guangxi's five major geographical divisions. Furthermore, ArcGIS spatial analysis tools were used to visualize and calculate the spatial distribution of traditional villages in Guangxi, as well as to summarize the characteristics of that distribution. Finally, data were stacked together, and the buffer analysis tool was used to explore the impact of natural factors, such as topography and geomorphology, and river and water systems, on the characteristics of the spatial distribution of traditional villages in Guangxi (Figure 2).

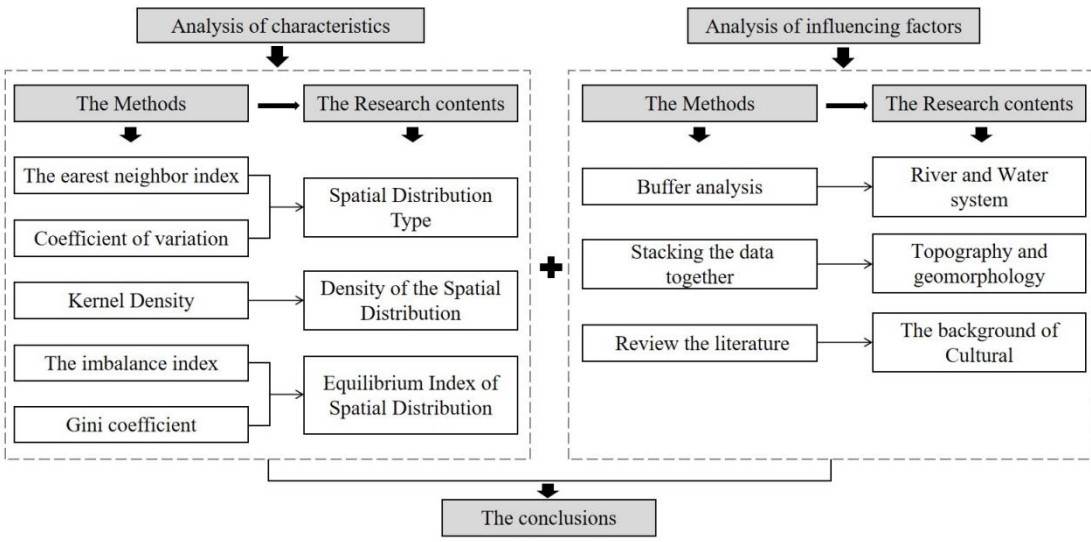

**Figure 2.** Research methods and technical route.

## 3. Analysis of Spatial Distribution Characteristics

*3.1. Types of Spatial Distribution*

The spatial locations of traditional villages are abstracted as point-like element data, and the distribution characteristics of point-like elements are usually classified into three types of spatial distribution, namely uniform, random, and cohesive [15]. The nearest neighbor index is used to analyze the types of spatial distribution of Guangxi's traditional villages, where this index can be used to determine the distribution state of point-like elements in the spatial system [14]. Its calculation formula is as follows:

$$R = \overline{r_1}/\overline{r_E} = 2\sqrt{D} \times \overline{r_1} \tag{1}$$

$$\overline{r_E} = \frac{1}{2\sqrt{n/A}} = \frac{1}{2\sqrt{D}} \tag{2}$$

where $R$ is the nearest neighbor index; $\overline{r_1}$ is the actual nearest neighbor distance; $\overline{r_E}$ is the theoretical nearest neighbor distance; $D$ denotes the point density; $A$ is the region's area and, in the text, the area of the Guangxi Autonomous Region $A$ = 240,200 km$^2$; and $n$ is the total number of traditional villages in the study area, that is, $n$ = 669. The point elements show the characteristics of uniform distribution and cohesive distribution, when $R > 1$ and $R < 1$, respectively.

Using the neighborhood analysis tool in an ArcGIS software platform's Spatial Analyst tool to visualize the analysis and calculate the correlation function, we obtain $\overline{r_1}$ = 6.72 km, $\overline{r_E}$ = 9.09 km, and $R$ = 0.74 < 1, indicating that the spatial distribution of traditional villages in Guangxi is a cohesive distribution.

It is still debatable whether the nearest neighbor index should be used to determine the type of distribution of point-like elements in a spatial system [16]. To prove further the applicability and accuracy of the nearest neighbor index's conclusion, the Spatial Analyst tool is used to create Voronoi polygons for the secondary calculation of the variation coefficient of traditional villages in Guangxi. The coefficient of variation measures an element's relative degree of spatial variation or its geographic phenomenon [17]. Its calculation formula is as follows:

$$CV = S/M \times 100\% \tag{3}$$

where $S$ represents the standard deviation of Voronoi polygons, $M$ is the mean of Voronoi polygon area, and $CV$ is the coefficient of variation. It is shown that when $CV > 64\%$, $33\% < CV < 64\%$, and $CV < 33\%$, the point elements are in a cohesive distribution, random distribution, and uniform distribution, respectively [18].

The coefficient of variation is calculated using the point elements of 669 traditional villages in Guangxi as the meta-unit to generate 669 Voronoi polygons, as shown in Figure 3, and the calculation results show that $S$ = 776.56 km$^2$, $M$ = 426.24 km$^2$, and $CV$ = 182.19%. The $CV$ value of the coefficient of variation is greater than 64%, verifying that the type of spatial distribution of traditional villages in Guangxi is cohesive, and that the judgment of the method of the nearest neighbor index is correct.

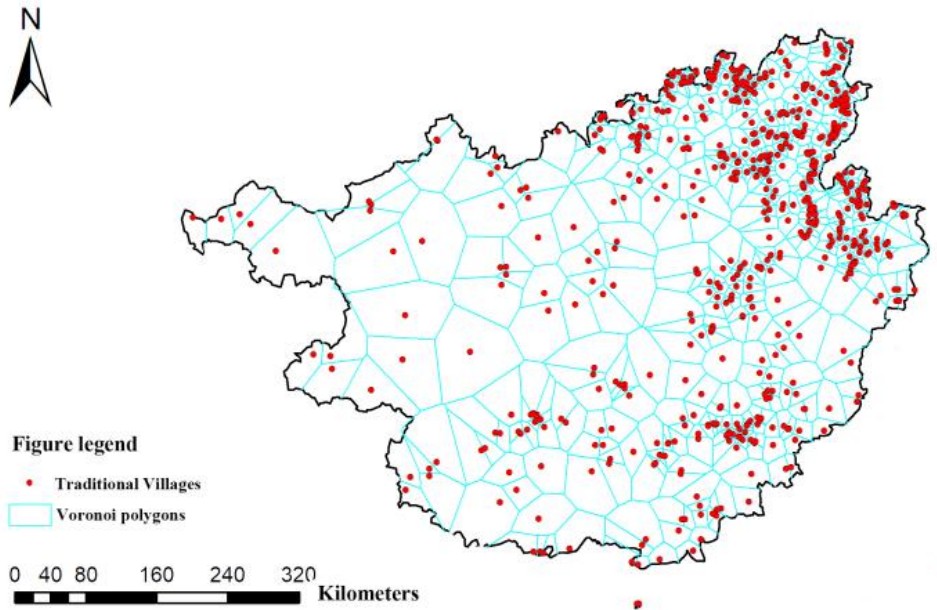

**Figure 3.** Voronoi polygon distribution of traditional villages in Guangxi.

### 3.2. Density of the Spatial Distribution

The kernel density tool in the ArcGIS software is used to analyze and process 669 traditional villages in Guangxi, and the kernel density distribution pattern of traditional villages

in Guangxi is derived, as shown in Figure 4. Figure 4 depicts the distribution pattern of traditional villages in Guangxi as "one main and two vices" of the multi-core cluster in general. On the one hand, "one main" refers to Guilin City and the northern areas of both Hezhou and Liuzhou, which formed the main and largest gathering area of the core of traditional villages; Guilin City, in particular, has the highest value of kernel density, totaling 271, and accounts for 40.5% of all traditional villages in Guangxi. On the other hand, "two vices" refers to the northern area of Yulin City and the northeastern part of Laibin City, which formed the sub-center of two gathering areas of traditional villages, totaling 72, and accounting for 11% of all traditional villages in that area. Other municipalities sporadically distribute fewer traditional villages within their boundaries. Thus, it can be observed that the distribution density of traditional villages in Guangxi reflects the region's uneven distribution characteristics.

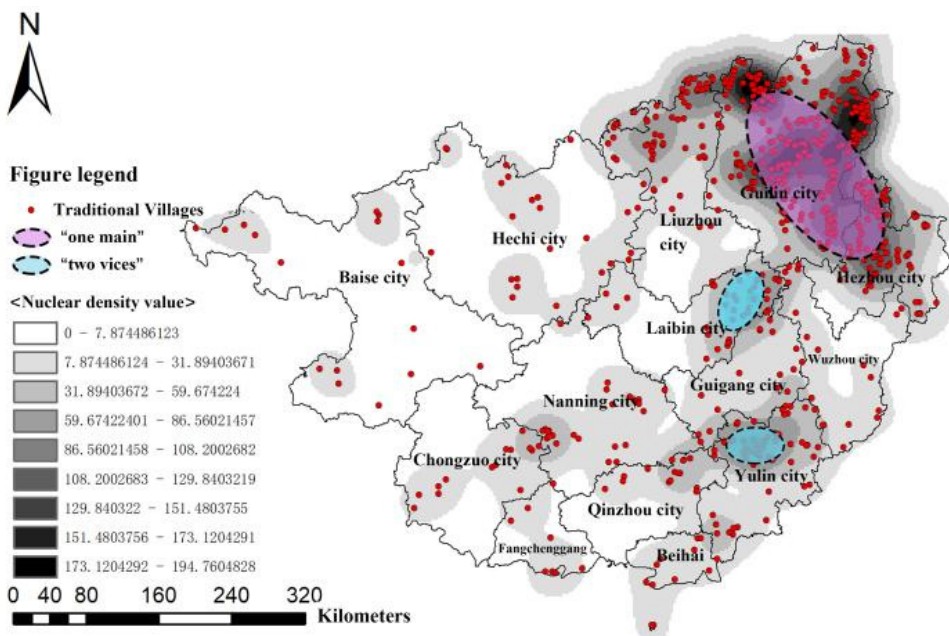

**Figure 4.** Distribution pattern of the kernel density of traditional villages in Guangxi.

*3.3. Analysis of the Equilibrium Index of Spatial Distribution*

3.3.1. Analysis of the Overall Degree of Equilibrium

The degree of equilibrium analysis is primarily a measure of whether the object of study is distributed at equilibrium across the entire domain, which can be accomplished by using the imbalance index method [18,19], whose formula is as follows:

$$S = \frac{\sum_{i=1}^{n} Y_i - 50(n+1)}{100n - 50(n+1)} \tag{4}$$

where $S$ is the imbalance index, and the value is between 0 and 1; $n$ is the number of municipalities; $Y_i$ is the cumulative percentage of the $i$-th position of the traditional villages in the total area in descending order, and the measured results are shown in Table 1. When $S = 0$, the objects of research are distributed evenly across each district; when $S = 1$, the objects of research are concentrated in a single district. The imbalance index $S = 0.56$ of Guangxi's traditional villages distribution is calculated, and the imbalance index is greater than the median value of 0.5, indicating that the spatial distribution of traditional villages in Guangxi is unbalanced within the municipal area.

**Table 1.** Statistics of traditional villages in cities of the Guangxi Zhuang Autonomous Region.

| City Name | Number of Traditional Villages/pc | Ranking | Percentage | Cumulative Percentage |
|---|---|---|---|---|
| Guilin | 271 | 1 | 40.5% | 40.5% |
| Hezhou | 86 | 2 | 12.9% | 53.4% |
| Liuzhou | 63 | 3 | 9.4% | 62.8% |
| Yulin | 58 | 4 | 8.7% | 71.5% |
| Laibin | 41 | 5 | 6.1% | 77.6% |
| Nanning | 25 | 6 | 3.7% | 81.3% |
| Hechi | 25 | 7 | 3.7% | 85% |
| Wuzhou | 20 | 8 | 3.0% | 88% |
| Guigang | 16 | 9 | 2.4% | 90.4% |
| Baise | 16 | 10 | 2.4% | 92.8% |
| Qinzhou | 15 | 11 | 2.2% | 95% |
| Chongzuo | 14 | 12 | 2.1% | 97.1% |
| Beihai | 12 | 13 | 1.8% | 98.9% |
| Fangchenggang | 7 | 14 | 1.1% | 100% |
| Total | 669 | | | |

### 3.3.2. Municipal Distribution Characteristics

The ArcGIS software is used to visualize the distribution of traditional villages in Guangxi, and the distribution of 14 municipalities of traditional villages in Guangxi is derived, as shown in Figure 5. Figure 5 shows that traditional villages in Guangxi are unevenly distributed at the municipal level. When combined with the values in Table 1, the municipal area with the darkest color block is Guilin City with 271, followed by the cities of Hezhou, Liuzhou, and Yulin with 86, 63, and 58, respectively. Western areas, such as the cities of Baise, Chongzuo, and Fangchenggang, have fewer distributed traditional villages, with only seven in Fangchenggang City. The distribution in municipal characteristics of traditional villages in Guangxi is such that the number of traditional villages distributed in the mountainous areas near the northeast of the Yuechengling mountain range, Tianping mountain range, and Jiuwan mountain range is greater than that in the central basin-like flat area. In contrast, the number of traditional villages distributed in the northwest Duyang mountain range area and the southern plain area of Qinzhou is the lowest.

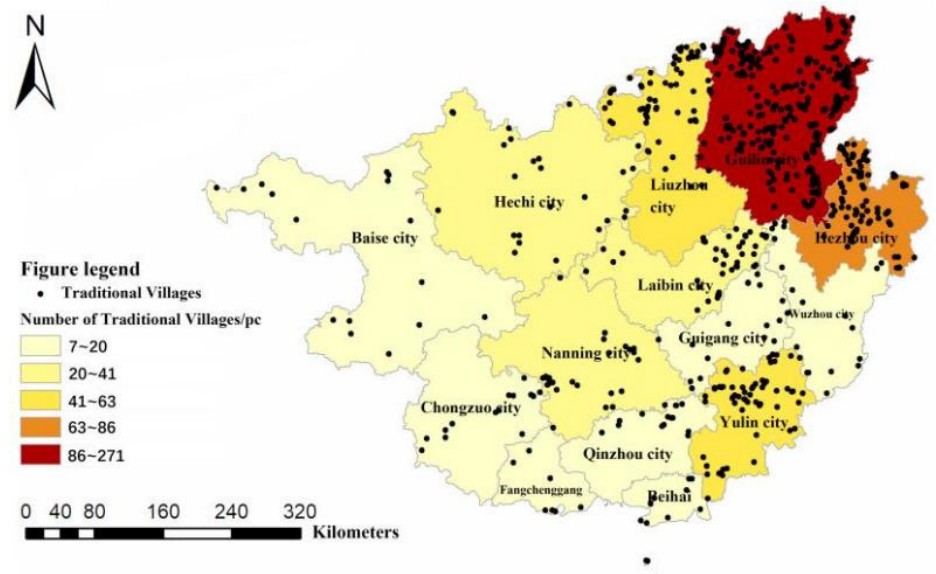

**Figure 5.** Urban distribution map of traditional villages in Guangxi.

### 3.3.3. Geographical Districts Distribution Characteristics

Guangxi was divided into five major geographic districts, according to the Guidelines for the Control and Management of the Characteristic Style of Agricultural Houses in Guangxi Zhuang Autonomous Region [20], which were introduced in March 2021, and these districts are as follows: Guibei, Guixi, Guizhong, Guinan, and Guidong. The 14 municipalities distributed in different geographic districts differed in various aspects, such as geographic environment, ethnic characteristics, tourism economy, and residents' living habits. Guibei District includes Guilin City, Longsheng, Lingchuan, Yongfu, Ziyuan, Xingan, Yangshuo, Lipu, Quanzhou, Guanyang, Gongcheng, Pingle, 13 counties, Hezhou City, Zhaoping, Zhongshan, Fuchuan County, Rongan County, Rongshui County, and Sanjiang County of Liuzhou City. Guixi District includes Baise City, Hechi City, and the counties (cities) within that jurisdiction. Guizhong District includes Nanning City, Chongzuo City, Laibin City, Liuzhou City, and the counties (cities) within that jurisdiction. Guinan District includrd Beihai City, Qinzhou City, Fangchenggang City, and the counties under their jurisdiction. Guidong District includes Wuzhou City, Yulin City, Guigang City, and the counties under their jurisdiction. The distribution of traditional villages in Guangxi in the five major geographic districts was statistically analyzed (e.g., Table 2). It can be concluded through numerical analysis that the distribution of traditional villages in the five major geographic districts of Guangxi shows a concentrated distribution, with Guibei District being the most concentrated area, accounting for 53.4% of the total distribution, followed by the northern regions of the Guizhong and Guidong Districts, which accounted for 21.3% and 14.1%, respectively. Notably, the other geographic districts are less distributed.

**Table 2.** Distribution statistics of traditional villages in five geographical districts of Guangxi.

| District | Total Number/pc | Percentage/% | Cumulative Percentage/% |
|---|---|---|---|
| Guibei District | 357 | 53.4 | 53.4 |
| Guizhong District | 143 | 21.3 | 74.7 |
| Guidong District | 94 | 14.1 | 88.8 |
| Guixi District | 41 | 6.1 | 94.9 |
| Guinan District | 34 | 5.1 | 100 |

The Gini coefficient can be used in geography to investigate the characteristics of discrete areas' spatial distributions and to compare the differences in the spatial distribution of regional geographic elements [21]. In this study, the Gini coefficient is used to verify the correctness of the values derived from Table 2, which are about the spatial distribution of traditional villages in Guangxi's five major geographic districts, and to increase the scientificity of the analysis results. The Gini coefficient $G$ is calculated using the following formula:

$$G = \frac{-\sum_{i=1}^{n} P_i ln P_i}{l_n N} \tag{5}$$

where $G$ is the Gini coefficient; $P_i$ is the proportion of the number of traditional villages in the $i$-th districts to the total number; $n$ is the number of districts and, in this study, $n = 5$. The value of $G$ ranges from 0 to 1; the closer the value of $G$ is to 1, the more concentrated and unbalanced the distribution is. The Gini coefficient of traditional villages in Guangxi's five geographical districts is calculated to be $G = 0.79$, indicating that the distribution of traditional villages in these districts is concentrated and unbalanced, and confirming that the results judged by the values in Table 2 are correct.

## 4. Analysis of Factors Affecting the Spatial Distribution of Traditional Villages in Guangxi

*4.1. Topography and Geomorphology*

Natural topography is an essential factor affecting the distribution of traditional villages during the generation and evolution of villages. Guangxi's topography and geomorphology are generally mountainous, with hilly basin landforms surrounded by many mountains, and the area is vast. The central and southern parts are mostly flat and basin-like, formed by karst landforms, with topographic features of high north and low south, owing to the northwestern part's connection with the Yunnan–Guizhou plateau and distribution by the Phoenix mountain range and the Duyang mountain range, and the northeastern part's distribution by the Yuechengling mountain range and the Tianping mountain range, with the highest point of the northeastern mountains being located in the Maoer mountains [22]; the southeastern and southern parts are the Yunkai mountains, the Dalong mountains, the Goulou mountains, and the Shiwan mountains, among others.

Figure 6 depicts a map of the digital elevation model (DEM) of the Guangxi topography coupled with 669 traditional villages using ArcGIS software. Figure 6 shows that traditional village clusters in Guangxi are mostly distributed in the low-lying areas on both sides of the mountain range, with only a few settlements distributed in the mountainous and plain areas. Using the "Extract Values to Points" in the Spatial Analyst tool to extract the information on DEM elevation of the corresponding point elements of traditional villages, it was found that approximately 62% of the traditional villages were distributed in the low-lying areas on both sides of the mountains below 300 m in elevation, including the core group of traditional villages in central Guilin and northern Hezhou, and the villages in the basin area south of the Darong Mountains and east of the Liouwan Mountains.

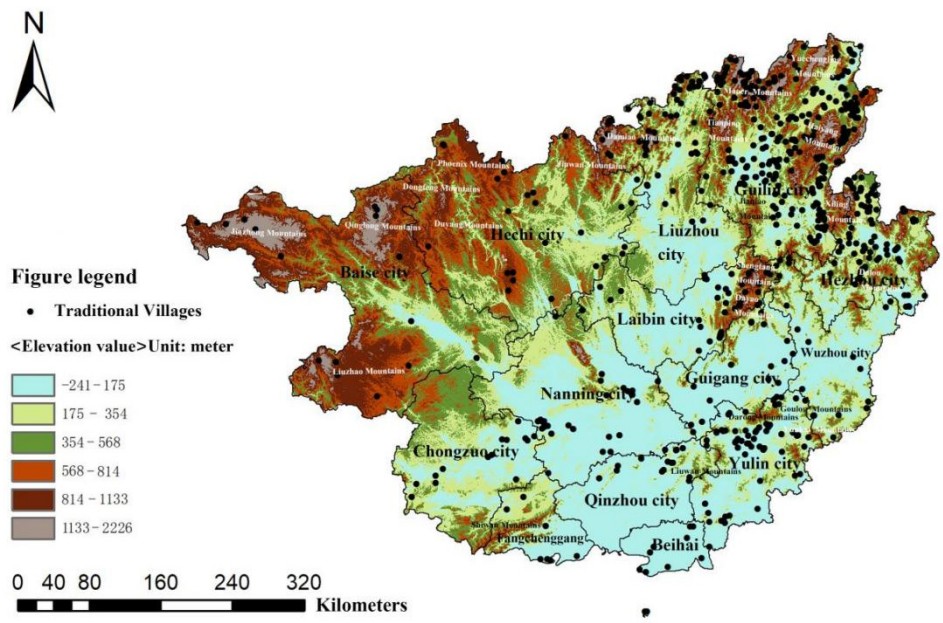

**Figure 6.** DEM elevation map and distribution of traditional villages in Guangxi.

The density of traditional villages in the northeastern Yuechengling Mountains, Tianping Mountains, and Jiuwan Mountains is significantly higher than that of the northwestern Duyang Mountains. Furthermore, the traditional villages in the northeastern mountainous areas are mostly clustered and distributed along the river valley areas and mountain margins on both sides of the mountains and are mainly influenced by the topography and are predominantly built near mountains and along the water. According to Figure 6, the gathering areas of traditional villages in northern Guilin and northeastern Laibin are mainly distributed in mountainous areas with elevation values ranging from 814 to 1133 m, and some traditional villages are distributed in mountainous areas ranging from 1133 to

2226 m, mainly containing the Yuechengling mountain range and Shengtang mountains, accounting for approximately 14.1% of the total number.

These traditional village clusters were built or distributed directly on the mountains. Owing to the treacherous terrain in mountainous areas, the relatively closed and inconvenient traffic, and the relatively slow urbanization process, they can be preserved in large numbers. Simultaneously, the rich forest resources can form a more independent geographical environment and villagers' living circle, allowing the original historical appearance of traditional ancient villages to be stably preserved.

On the contrary, the number of traditional villages in the flat areas of the country's central and southern parts is relatively sparse, owing to the influence of migrant culture in the plains, which makes people more mobile, and means that the villages' appearance and cultures are easily influenced by foreign cultures.

*4.2. River and Water Systems*

Choosing to live near water is typical ancient wisdom, and it is a common principle for forming primitive ancient villages; however, flood prevention and disaster resistance are also important factors to consider in village site selection. Guangxi has many rivers, and the water network distribution is dense. The main water system in Guangxi is the Xijiang river system in the Pearl River basin, and the main stream is from the Nanpanjiang River–Hongshui River–Qianjiang River–Xunjiang River–Xijiang River, which crosses the entire territory of the Guangxi Zhuang Autonomous Region from northwest to east along the terrain. The Yangtze River basin, whose main streams are the Xiangjiang and Zijiang River systems, is predominantly distributed in the northeastern part of Guangxi. The Ling Canal, built during the Qin Dynasty, is the main channel connecting the Yangtze and Pearl River basins; the Red River is Guangxi's mother river, and more water systems in Guangxi's south are injected into the Beibu Gulf.

The buffer of the neighborhood analysis tool was used to analyze the influence of river systems on the spatial distribution of traditional villages in Guangxi, and buffer zones of 1 km, 3 km, and 5 km away from rivers were established in the entire Guangxi area as shown in Figure 7. According to calculations, approximately 547 traditional villages are distributed on both sides of the 5 km buffer zone along the rivers, accounting for approximately 81.8% of the total. Thus, most traditional villages in Guangxi can be found along the river systems, the villages are distributed along the direction of the water system, and their locations are closely related to the river systems. Among them, the gathering area of traditional villages in Guilin's north is mainly distributed around the Li Jiang and Xiang Iiang; the gathering area of traditional villages in Liuzhou's north is mainly distributed around the Rong Jiang; the gathering area of traditional villages in Yulin's north is mainly around the Nanliuhe River; the gathering area of traditional villages in Laibin's northeast runs through the Hongshui River, Qianjiang River, and Liujiang River basin. In addition, Figure 7 shows that some traditional villages are located outside the 5 km buffer zone from rivers and are relatively far away from rivers, for example, on the southeast side of the Liujiang River basin. Traditional villages in this area are mostly located away from rivers that frequently flood in order to deal with the problem of flood prevention and disaster prevention. In the case of Liujiang, data show that major floods have occurred more frequently and that disasters have been severe over the last 20 years, from 1998 to the present [23]. Therefore, it is possible to conclude that river systems play an essential role in determining the location of traditional villages in Guangxi.

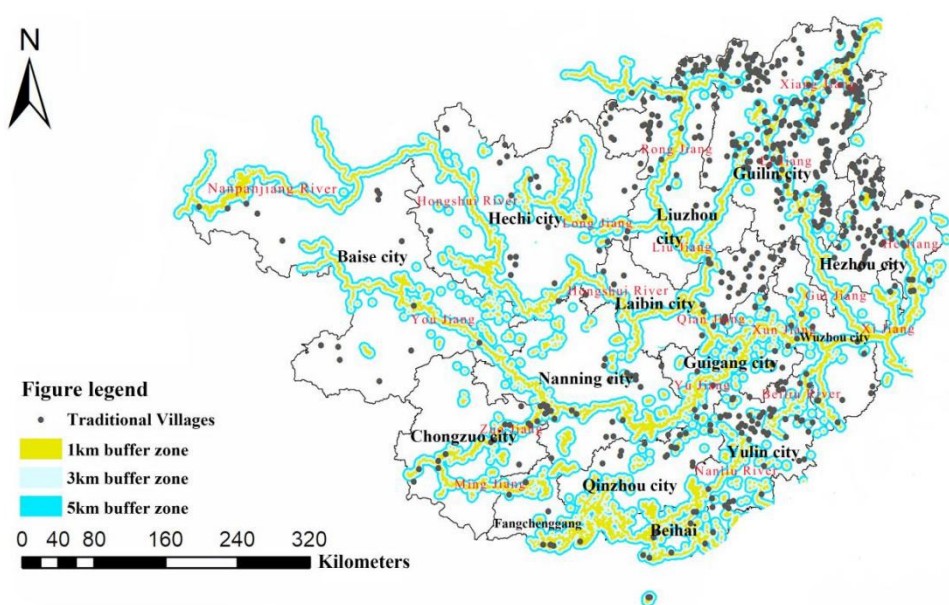

**Figure 7.** Water systems and distribution of traditional villages in Guangxi.

## 5. Conclusions

This study uses 669 traditional villages in Guangxi as the research object, and conducts research on the characteristics of traditional villages' spatial distribution using the nearest neighbor point index, coefficient of variation, kernel density, Gini coefficient, and imbalance index, and draws the following conclusions:

(1) Through the method of nearest point index and coefficient of variation, the spatial distribution of traditional villages in Guangxi is cohesive, and the analysis of kernel density shows that the spatial distribution of traditional villages in Guangxi is a multi-core agglomeration pattern of "one main and two vices," with Guilin, northern Hezhou, and northern Liuzhou forming the main core agglomeration area of traditional villages, and northern Yulin and northeastern Laibin being the "two vices" of agglomeration centers of traditional villages;

(2) The Gini coefficient method reveals that traditional villages are primarily concentrated in the Guibei District, followed by the central and northern regions of the Guizhong District, and the Guidong District, with unbalanced distribution; the unbalanced index demonstrates that traditional villages are mainly distributed in Guilin, Hezhou, Liuzhou, and Yulin, with Guilin forming the highest density of traditional villages. This imbalance reflects the fact that villages development is a spontaneous construction process with prominent self-organization characteristics;

(3) The influencing factors of the characteristics of the spatial distribution of traditional villages in Guangxi are mainly affected by environmental factors, such as the intersection of topography and geomorphology, and river and water systems. Among these, closed traffic, dangerous terrain, and fertile water resources are important factors for the location, as well as the preservation and development of, traditional villages; furthermore, they are important reasons why traditional villages can be developed in clusters and aggregations.

**Author Contributions:** Conceptualization, M.L. and W.O.; methodology, M.L.; software, M.L.; validation, M.L.; formal analysis, M.L.; investigation, M.L.; resources, W.O.; data curation, W.O.; writing—original draft preparation, M.L.; writing—review and editing, M.L.; visualization, M.L.; supervision, M.L.; project administration, D.Z.; funding acquisition, D.Z. All authors have read and agreed to the published version of the manuscript.

**Funding:** This work is supported by two National Natural Science Foundation of China projects [grant numbers 51878021, 51938002].

**Informed Consent Statement:** Informed consent was obtained from all subjects involved in the study.

**Data Availability Statement:** Not applicable.

**Conflicts of Interest:** We declare that we have no financial and personal relationships with other people or organizations that can interfere with our study.

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
