# Peer review of "Spatial Distribution Characteristics and Influencing Factors of Traditional Villages in Guangxi Zhuang Autonomous Region"

_sustainability, doi:10.3390/su15010632_

Round 1
Reviewer 1 Report
The subject of the paper is interesting in that it introduces the issue of protection of traditional settlement and endavours to utilize quantitative methods to examine this issue.
However the paper appears to utilize some GIS and statistical analysis to cope with an issue that is not of a quantitative character.
First of all the objective of the paper is not declared from the begining (summary and introduction), instead it appears that the paper aims to show how the specific techinques and tools can be used to analyze characteristics of settlements distribution . Of course, this is another issue which can apply to all settlrements, not only to traditional ones.
Besides no theoretical framework is presented to interprete how settlements in general and in this area in particular are created and spatially distributed and how this relates to their traditional character.
Conclusions repeat exixting knowledge, e.g. that rivers influence settlements location and only few findings is added to existing knowledge
The paper should be revised drastically. The aim should be declared from the begining and be connected with a planning disicipline such as settlements protection through theoretical and analytical tools. Description of tools should be limited to necessary, in seeking to higlight the key arguements that now are missing.
The issues addressed should be put in their theoretical framework so that details about measurements and quantitative tools to support the key arguments instead of being the subject of the paper.
It is crucial some basic theories of settlements organization to be refered to, as is for instance Chrystaller's theory.
Besides it is important to accentuate the relationship of the analysis with settlements' protection.
As revealed from the title, the paper should extend its scope beyond the specifc province it examines and try to put it in a more general freamework, so that some lessons to be learnt for policy making.
Author Response
1. The author revised the abstract, put forward the research objective directly, and weakened the research method.
2. In the introduction, a theoretical framework is added to explain the general settlement, especially how the settlements in this region are established and distributed in space, as well as the relationship between this and their traditional characteristics.
3. Reorganize the repeated part of the conclusion.
Reviewer 2 Report
This study explores the spatial distribution of traditional villages in Guangxi. It is interesting, and the paper is well-structured. I suggest accepting this paper for publication.
Author Response
No comments, no need to modify, thank you.
Reviewer 3 Report
Spatial Distribution Characteristics and Influencing Factors of Traditional Villages in Guangxi Zhuang Autonomous Region
In this manuscript, the authors analyze the correlation between topography and river systems and the spatial location and distribution density of traditional villages in Guangxi revealing their influence on the distribution of spatial villages. Traditional villages are geographically concentrated in the Yuecheng Ling mountain range and Guibei’s surrounding flat areas, Guizhong’s Shengtang Mountain range, and the Guidong’s alluvial river plains. Traditional villages are more prevalent in mountainous areas, and their construction and development take the water resources of rivers and flood protection into account.
After I read this manuscript, I realize the manuscript has some good points such as the idea behind, the structure of the paper, and the interest of some particular audience. I have realized some limitations. I can recommend it for publication after the following considerations are taken into account:
1. The abstract needs to be restructured. The structure and the logic succession between ideas is quite random and not showing any coherent structure at all. Some aspects related to:
· What do the authors mean with “Traditional villages are (…) a non-renewable resource”. Can we talk about renewable resources when we talk about some very abstract ideas behind traditional villages (iincluding culture, architecture, traditions, etc). I do not see clear this argument.
· The connection between sentences is not logical and subsequent at all. For example, the authors argue “Traditional villages are an important part of Chinese farming civilization and a non-renewable resource; therefore, research…”. Therefore? The first sentence is pretty generic and abstract. What are the reasons for making huge investments in the preservation of traditional villages in China? That is the question.
· And suddenly the tools and methodology implemented: “Using the ArcGIS software platform, the nearest neighbor index and coefficient of variation and Kernel Density tools. (…) the imbalance index method and the Gini coefficient are used to describe the equilibrium index of the distribution of traditional villages in municipalities and geographical subdivisions.” But the problem was not described before. Also the authors introduce the tools, but they argue only some lines ahead why they are using them.
· The authors argue: “the correlation between topography and river systems and the spatial location and distribution density of traditional villages in Guangxi reveals their influence on the distribution of spatial villages”. This is barely coherent. Why do the authors include so many factors at the same time? Why do they not talk about the correlation between physical/environmental and human factors?
· What do the authors mean here with “vices”? “They demonstrates that Guangxi’s traditional villages are spatially distributed with “one main and two vices.”
· And at the end of the abstract, the authors argue: “This study provides guidance for the protection and utilization of traditional villages in the Guangxi area.” How they do? How to know where these historical towns are located help to their preservation? Do you mean cataloging them?
2. Another aspect I am missing in the abstract, but also in the paper, is the importance of this study from an international perspective. Why are the authors focusing on that study area? How their results can be extrapolated to other world regions?
3. In the introduction, the authors argue: “Traditional villages are important carriers of Chinese farming culture and cultural heritage resources, with significantly high historical, cultural, scientific, artistic, economic, and social values [1], which are irreproducible. Certain issues have arisen in recent years due to a lack of attention to traditional villages and inappropriate protection and utilization, including the destruction of traditional texture, the destruction of traditional architectural style, the disappearance of traditional regional cultural characteristics, and other problems.” I am missing references related to official data published by international institutions such UNESCO, for example. Also, this institution is talking about tangible and intangible heritage, that is something that the authors mention in some way
4. After that, the authors argue: “Since 2012, five batches of traditional Chinese villages have been selected, including 6,819 villages and towns. Concurrently, several provinces, cities, and districts have accelerated the assessment and protection of traditional villages at different levels.” Where these batches were selected?
5. The authors argue: “This mainly involves interdisciplinary fields, such as architecture, sociology, history, planning, and psychology, which have become critical in supporting the field of traditional village protection. Domestic scholars’ research on traditional villages majorly focuses on traditional villages’ connotation and culture [1,2], their spatial form [3,4], their spatial structure [5-7], their public space [8], and their tourism development [9,10].” After this, I have some suggestions that the authors must consider:
a. I am missing relevant references to one of the most important focuses for the preservation of these historical villages. Experts from different fields of knowledge work on “digital preservation” by implementing technologies for surveying the most relevant architectural structures. I can recommend you the studies from Owda et al. (“Methodology for digital preservation of the cultural and patrimonial heritage: Generation of a 3D model of the church St. Peter and Paul (Calw, Germany)”) or Balsa and Fritsch (“Generation of visually aesthetic and detailed 3D models of historical cities by using laser scanning and digital photogrammetry” and “Generation of 3D/4D photorealistic building models. The testbed area for 4D Cultural Heritage World (4D-CHW) project”)
b. This list of studies is unstructured and this paragraph must be extended by including a longer discussion about these studies and the importance of the different perspectives considered.
c. Most of the studies that the authors mention are from China and/or have a local implementation. I recommend them to include a broad list of relevant international studies.
6. The authors argue also in the introduction: “The research on the spatial distribution of traditional villages and its influencing factors using geography combined with big data analysis has become a new hot spot with the gradual popularization of the application of big data and information technology in the field of urban and rural planning and architecture [11-14]. Furthermore, most current related studies are based on core density, geographic agglomeration, and other perspectives in order to cut into the multiperspective analysis”. Some relevant references to studies talking about the complex systems behind spatial patterns related to population distribution over time. Paul Krugman studied this pattern for Europe in the middle age that was based on rural societies where villages emerged for trading products. Nowadays, some studies analyze how the patterns are changed based on the urban economics factors such the study published by Balsa, Morales and Pentland (“Mapping population dynamics at local scales using spatial networks”) where random traditional villages are experiencing urban growths imitating what is happening at larger scales, while the rest of the population settlements are declining population over time. For this, I recommend you in the discussion of your results to evaluate what is happening with these towns nowadays and why the current trends in China (based on fast urbanization in some regions) is leading to the problems of depopulation and deterioration of these traditional villages.
7. Figures 1, 3,4 and 5 are pretty small. Please make them bigger and increase the resolution.
8. Why the caption of the figure 5 is located over the figure?
9. In Figure 6 why the first interval of the DEM ranges from -241 to +175. I would suggest to use one interval just for the values below 0
1 The list of references is very limited to local studies focused on China. However, I am expecting to project this study to a broader audience and to include studies that are more international.
1 I have also observed some limitations with some expressions such, for example: “…spatial location and distribution density of traditional villages in Guangxi revealing…”. I would suggest reviewing the whole manuscript with some language service.
Author Response
1. Reorganize the content of the abstract.
2. The method of the research can be extrapolated to other world regions.
3.The rapid disappearance of traditional villages is supported by official data。
4.These numbers were obtained by consulting traditional Chinese village websites.
5.This paper focuses more on the use of big data technology, but does not focus on digital protection, so it does not cover the relevant papers on architectural structure design of digital protection.
6.Add references to relevant foreign literatures
7. Figures 1, 3,4 and 5 are bigger and increase the resolution.
8.The caption of the figure 5 is located at the bottom the figure.
9.The first interval of the DEM ranges from -241 to +175 can analyze the lowest elevation.
10.The list of references is not limited to local studies focused on China. While include studies that are more international.
Reviewer 4 Report
This article addresses an issue which can be of great relevance for the future of Traditional village settlement system left in the country.
However, there are several shortcomings in the approach. A 'Conceptual Framework' was badly missed for understanding what the research question is and on what assumptions this study was carried out. As a result, it does not clearly spell out why is cohesiveness so important for spatial distribution of villages. Similarly, how concentration of traditional villages in 14 cities or 5 districts so important to analyze. Merely stating the descriptive statistics will not contribute to the domain knowledge.
In similar lines, why should we be analyzing the elevation or proximity to river or low lying areas - how are they conceptually related to decline in health of traditional villages? A better literature review is needed to explore the alternative causal explanations on life and death of traditional settlements. And if they do not exist, the authors must propose a model with distribution network/topographical parameters and empirically validate it.
Some other issues also need clarification. Why was spatial auto-correlation analysis (say Moran's I or Geary) not done. Is cohesiveness in spatial distribution extensively used for agro-ecosystems rather than human settlement systems? We need more justification for these methods used. Again why was concentration not estimated with 14 units rather that 5 districts? Moreover, a short note on the scale of units and their hierarchy needed some elaboration. Phrases like "14 municipalities of traditional villages..." create some confusion to audience who are not familiar with Chinese settlement system.
Lastly, If one is really interested in finding what spatial distribution or topographic features are crucial to the existence of traditional villages, one can do that more efficiently by longitudinal research design. A note of explanation will be helpful as why it stuck to cross-sectional approach.
And it would also be nice, if authors can explain why population size or size/value of agro-produce do not appear in the analytical framework.
Author Response
1.The 14 municipalities directly under the Central Government and the five districts analyzed and discussed the overall aggregation degree from different levels. The analysis of the 14 municipalities directly under the Central Government is shown above。
2.The development and evolution of traditional villages is mainly influenced by the natural geographical environment, especially the topography and landform and river system, which is consistent with the traditional Chinese feng shui thought.
3.This paper mainly considers the use of quantitative methods to analyze its spatial distribution characteristics, while the population size is more qualitative data, which is inconsistent with the starting point of this paper.
Round 2
Reviewer 1 Report
My previous review stressed that the paper should be drastically revised.
However this did not happen. On the contrary, the writers made some marginal changes, namely:
- minor restructure of the abstract
- insertion of 3-4 lines in the introductory part, which can not be considered as a theoretical framework.
Reviewer's recommendations to connect empirical data with theoretical models of settlement organization as well as with a settelments' protection are ingnored.
The subject of the paper remains focused on tools and measurements which should be clearly stated in the title and the (missing) theoretical part.
No reorganization of conclusions is evident
Changes in the paper are insufficient or are not clearly shown.
English language proofreading is strongly recommended
Author Response
Dear Professor,This paper is more considered from the perspective of quantitative analysis, is focused on tools and measurements.
Reviewer 4 Report
Comments made are addressed in the revision.
Author Response
Thank you very much for your correction and accept it humbly